# Adversarially Robust Learning via Entropic Regularization

**Gauri Jagatap** [1]  **Ameya Joshi** [1]  **Animesh Basak Chowdhury** [1]  **Siddharth Garg** [1]  **Chinmay Hegde** [1]

## Abstract

In this paper we propose a new family of algorithms, ATENT, for training adversarially robust deep neural networks. We formulate a new loss function that is equipped with an entropic regularization. Our loss considers the contribution of adversarial samples that are drawn from a specially designed distribution that assigns high probability to points with high loss and in the immediate neighborhood of training samples. ATENT achieves competitive (or better) performance in terms of robust classification accuracy as compared to several state-of-the-art robust learning approaches on benchmark datasets such as MNIST and CIFAR-10.

## 1. Introduction

Deep neural networks have led to significant breakthroughs in various fields, but have also been shown to be very susceptible to carefully designed "attacks" (Goodfellow et al., 2014b; Papernot et al., 2016) both on input data as well as network weights (Biggio et al., 2013).

Formally, the forward map between the inputs $x \in \mathbb{R}^d$ and outputs $y \in \{1, 2 \ldots m\}$ is modelled via a neural network as $y = f(w; x)$ where $w$ represents the set of trainable weight parameters. The collection of all labeled data $\{x_i, y_i\}$, $i = 1, \ldots, n$ can be represented as $X$ and $Y$. Then, the neural prediction $\hat{y}(x) = f(\hat{w}; x)$, can be very sensitive to changes in both $\hat{w}$ and $x$. For a bounded perturbation to a test image input (or to the neural network weights), $\hat{y}_i = f(\hat{w}; x_i + \delta_i)$ where $\delta_i$ represents the perturbation, the predicted label $\hat{y}_i$ can be made *arbitrarily* different from the true label $y_i$.

Typically, adversarial perturbations are constructed by maximizing the loss function within a neighborhood around the test point $x$ (Tramèr et al., 2017; Madry et al., 2018):

$$\bar{x}_{\text{worst}} = \arg \max_{\delta \in \Delta_p} L(f(\hat{w}; x + \delta), y) \qquad (1)$$

---
[1]New York University, USA. Correspondence to: Gauri Jagatap <gauri.jagatap@nyu.edu>.

*Accepted by the ICML 2021 workshop on A Blessing in Disguise: The Prospects and Perils of Adversarial Machine Learning.* Copyright 2021 by the author(s).

where $\hat{w}$ are the final weights of a pre-trained network. The perturbation set $\Delta_p$ is typically chosen to be an $\ell_p$-ball for some $p \in \{0, 1, 2, \infty\}$.

The existence of adversarial attacks motivates the need for a "defense" mechanism that makes the network under consideration more robust. We discuss several families of effective defenses. The first involves *adversarial training* (Madry et al., 2018). Here, a set of adversarial perturbations are constructed by solving a min-max objective of the form:

$$\hat{w} = \min_w \max_{\delta \in \Delta_p} \frac{1}{n} \sum_{i=1}^{n} L(f(w; x_i + \delta), y_i).$$

Wong & Kolter (2018) use a convex outer adversarial polytope as an upper bound for worst-case loss in robust training. (Tjeng et al., 2018) propose mixed-integer programming based certified training for piece-wise linear neural networks, (Gowal et al., 2019) use integer bound propagation, and (Lecuyer et al., 2019; Cohen et al., 2019; Salman et al., 2019b) show certified defenses via randomized smoothing. Due to space constraints we push discussion on prior work to appendix.

In this paper, we propose a new approach for training adversarially robust neural networks. The key conceptual ingredient underlying our approach is *entropic regularization*. Borrowing intuition from (Chaudhari et al., 2019), instead of the empirical risk (or its adversarial counterpart), our algorithm optimizes over a local entropy-regularized version of the empirical risk: $\hat{w} = \arg \min_w, \mathcal{L}_{DE}$

$$\mathcal{L}_{DE} = \int_{X'} \mathcal{L}(X'; Y, w) \left[ \frac{e^{\left( \mathcal{L}(X'; Y, w) - \frac{\gamma}{2} \|X - X'\|_p^p \right)}}{Z} \right] dX'. \qquad (2)$$

Intuitively, this new loss function can be viewed as the convolution of the empirical risk with a Gibbs-like distribution to sample points from the neighborhoods, $X'$, of the training data points $X$ that have high loss. Therefore, compared to adversarial training, we have replaced the inner maximization with an expected value with respect to a modified Gibbs measure which is matched to the geometry of the perturbation set. We use Stochastic Gradient Langevin Dynamics (Welling & Teh, 2011) to sample $X'$; in this manner,

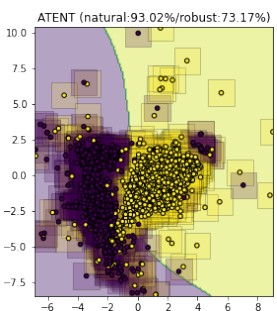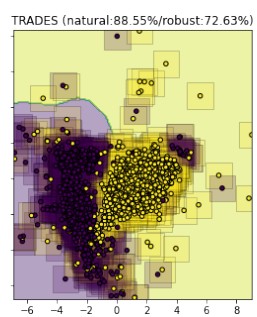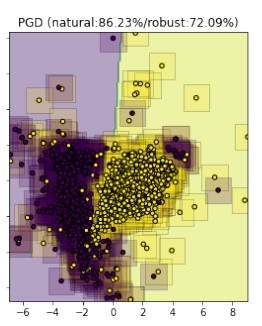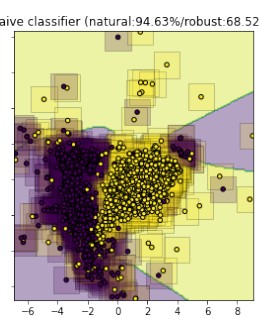

*Figure 1.* TSNE visualization of decision boundaries for a 3-layer neural network trained using different defenses; corresponding natural and robust test accuracies against $\ell_\infty$ attacks for classifying MNIST digits 5 and 8.

our approach blends in elements from adversarial training, randomized smoothing, and entropic regularization. We posit that the combination of these techniques will encourage a classifier to learn a better robust decision boundary as compared to prior art (see visualization in Fig.1). We name this entire procedure Adversarial Training with ENTropy or ATENT.

In this paper, we show that ATENT-trained networks provide improved (robust) test accuracy when compared to state of art defense approaches such as TRADES and MART. We also combine randomized smoothing with ATENT to show competitive performance with the smoothed version of TRADES. In particular, we are able to train an $\ell_\infty$-robust CIFAR-10 model to 57.23% accuracy at PGD attack level $\epsilon = 8/255$, which is higher than the latest benchmark defenses based on both adversarial training using early stopping (Salman et al., 2019a) (56.8%) as well as TRADES (56.6%) (Zhang et al., 2019b).

## 2. Problem Formulation

To model for adversarial perturbations in the samples, we design an augmented loss that regularizes the data space (data-space version of Entropy-SGD ((Chaudhari et al., 2019), recapped in Appendix C). Note that we only seek specific perturbations of data $x$ that *increase* the overall loss value of prediction. In order to formally motivate our approach, we first make some assumptions.

**Assumption 1.** *The distribution of possible adversarial perturbations follows:*

$$p(X'; X, Y, w, \gamma) = \qquad\qquad\qquad\qquad (3)$$

$$\begin{cases} Z_{X,w,\gamma}^{-1} e^{\mathcal{L}(X';Y,w) - \frac{\gamma}{2}\|X'-X\|_F^2} & if \quad \mathcal{L}(X';Y,w) \leq R \\ 0 & if \quad \mathcal{L}(X';Y,w) > R \end{cases}$$

*where $Z_{X,w,\gamma}$ is the partition function that normalizes the probability distribution.*

Intuitively, the neural network is more likely to "see" perturbed examples from the adversary corresponding to higher

loss values. The parameter $R$ is chosen to ensure that the integral of the probability curve is bounded. Here $\gamma$ controls the penalty of the distance of the adversary from true data $X$; if $\gamma \to \infty$, the sampling is sharp, i.e. $p(X' = X; X, Y, w, \gamma) = 1$ and $p(X' \neq X; X, Y, w, \gamma) = 0$, which is the same as sampling only the standard loss $\mathcal{L}$, meanwhile $\gamma \to 0$ corresponds to a uniform contribution from all possible data points in the loss manifold.

We design a new loss function $\mathcal{L}_{DE}(w; X, Y, \gamma)$ which incorporates the probabilistic formulation in Assumption 1:

$$\mathcal{L}_{DE} = \int_{X'} \mathcal{L}(X'; Y, w) p(X'; X, Y, w, \gamma) dX' \quad (4)$$

our new objective is to minimize this augmented objective function $\mathcal{L}_{DE}(w; X, Y, \gamma)$, which resembles expected value of the standard loss function sampled according to a distribution that (i) penalizes points further away from the true training data (ii) boosts data points which correspond to high loss values. This sampling process (Fig. 3), as well as theoretical properties of our augmented loss function (Lemma C.1) are described in detail in supplement (Appendix C).

If gradient descent is used to minimize the loss in Eq. 4, the gradient update $\nabla_w \mathcal{L}_{de}(w; X, Y, \gamma)$ corresponding to the augmented loss function can be computed as follows

$$\nabla_w \mathcal{L}_{de} = \nabla_w \mathbb{E}_{X' \sim p(X'; X, Y, w, \gamma)}[\mathcal{L}(X'; Y, w)] \quad (5)$$

Correspondingly, the weights of the network, when trained using gradient descent, using Eq. 5, can be updated as

$$w^+ = w - \eta \nabla_w \mathbb{E}_{X' \sim p(X'; X, Y, w, \gamma)}[\mathcal{L}(X'; Y, w)] \quad (6)$$

where $\eta$ is the step size. The expectation in Eq. 5 is carried out over the probability distribution of data samples $X'$ as defined in Assumption 1.

The expectation in Eq. 5 is computationally intractable to optimize (or evaluate). However, using the Euler discretization of the Langevin Stochastic Differential Equation (Welling

& Teh, 2011), it can be approximated well. Samples can be generated from $p(X')$ as:

$$X'^{k+1} = X'^k + \eta' \nabla_{X'} \log p(X'^t) + \sqrt{2\eta'} \varepsilon \mathcal{N}(0, \mathbb{I}) \quad (7)$$

where $\eta'$ is the step size for Langevin sampling, $\varepsilon$ is a scaling factor that controls additive noise. In Langevin dynamics, when one considers a starting point of $X'^0$ then the procedure above yields samples $X'^1 \ldots X'^t$ that follow the distribution $p(X')$.

Observe that $X'$ and $X$ have the same dimensions and the gradient term in the above equation needs to be computed over $n$, $d$-dimensional data points. In practice this can be computationally expensive. Therefore, in practice we use a stochastic variant of this update rule, which considers mini-batches of training data instead (training data is segmented into $J$ batches $[X_{B_1}, X_{B_2} \ldots X_{B_J}]$). The update rule is

$$\frac{X'^{k+1} - X'^k}{\eta'} = \quad (8)$$

$$\nabla_{X'^k} \mathcal{L}(X'^k; Y, w) + \gamma(X - X'^k) + \sqrt{\frac{2\varepsilon^2}{\eta'}} \mathcal{N}(0, \mathbb{I})$$

where we have incorporated $Z_{X,w\gamma}$ in the step size $\eta'$. Note that as the number of updates $k \to \infty$, the estimates from the procedure in Eq. 8 converge to samples from the true distribution. $p(X'; X, Y, w, \gamma)$. We then want to estimate $\nabla_w \mathcal{L}_{de}(w; X, Y, \gamma) = \nabla_w \mathbb{E}_{X' \sim p(X')} \left[ \mathcal{L}(w; X', Y, \gamma) \right]$ using the samples obtained from the above iterative procedure.

This discussion effectively leads to the algorithm shown in Algorithm 1 in Appendix C, which we refer to as Adversarial Training using Entropy (or ATENT), designed for $\ell_2$ attacks.

In Appendix B we discuss analytical comparison of ATENT to PGD-AT and randomized smoothing.

***Extension to defense against $\ell_\infty$-attacks:*** It is evident that due to the isotropic structure of the Gibbs measure around each data point, Algorithm 1, $\ell_2$-ATENT is best suited for $\ell_2$ attacks. However this may not necessarily translate to robustness against $\ell_\infty$ attacks.

**Assumption 2.** *We consider a modified distribution to account for robustness against $\ell_\infty$ type attacks:*

$$p(X'; X, Y, w, \gamma) = \quad (9)$$
$$\begin{cases} Z_{X,w,\gamma}^{-1} e^{\left( \mathcal{L}(X'; Y, w) - \frac{\gamma}{2} \|X' - X\|_\infty \right)} & \text{if} \quad \mathcal{L}(X'; Y, w) \leq R \\ 0 & \text{if} \quad \mathcal{L}(X'; Y, w) > R \end{cases}$$

*where $\| \cdot \|_\infty$ is the $\ell_\infty$ norm on the vectorization of its argument and $Z_{X,w,\gamma}$ normalizes the probability.*

The corresponding Data Entropy Loss for $\ell_\infty$ defenses is:

$$\mathcal{L}_{DE,\infty}(w; X, Y) =$$
$$Z_{X,w,\gamma}^{-1} \int_{X'} \mathcal{L}(X'; Y, w) e^{\left( \mathcal{L}(X'; Y, w) - \frac{\gamma}{2} \|X - X'\|_\infty \right)} dX'$$

and the SGD update to minimize this loss becomes:

$$\nabla_w \mathcal{L}_{DE,\infty}(w; X, Y) = \nabla_w \mathbb{E}_{X' \sim p(X')} \left[ \mathcal{L}(w; X', Y) \right]$$
$$\implies w^+ = w - \eta \nabla_w \mathcal{L}_{DE,\infty}(w; X, Y)$$

where the expectation over $p(X')$ is computed by using samples generated via Langevin Dynamics:

$$X'^{k+1} = X'^k + \eta' \nabla_{X'} \log p(X'^k) + \sqrt{2\eta'} \varepsilon \mathcal{N}(0, \mathbb{I})$$

Plugging in the distribution in Assumption 2 the update rule for sampling $X'$:

$$\frac{X'^{k+1} - X'^k}{\eta'} = \quad (10)$$

$$\nabla_{X'^k} L(X'^k; Y, w) + \gamma \text{sign}(X_i - X_i'^k) \cdot \mathbf{1} + \sqrt{\frac{2\varepsilon^2}{\eta'}} \mathcal{N}(0, \mathbb{I})$$

where $i = \arg\max_j |X_j - X_j'^k|$ and $j$ scans all elements of the tensors $X, X'^k$ and $\mathbf{1}_j = \delta_{i,j}$. The second term in the update rule navigates the updates $X'^{k+1}$ to lie in the immediate $\ell_\infty$ neighborhood of $X$. Note that this training process requires taking gradients of $\ell_\infty$ distance. In the update rule in Eq. 10, the gradient update only happens along one coordinate. In practice with this update rule, the algorithm fails to converge because typically a sizeable number of elements of $X' - X$ have a large magnitude.

Similar to the $\ell_\infty$ Carlini Wagner attack (Carlini & Wagner, 2017a), we replace the gradient update of the $\ell_\infty$ term, with a clipping based projection oracle. We design an accelerated version of the update rule in Eq. 10, in which we perform a clipping operation, i.e. an $\ell_\infty$ ball projection of the form:

$$X'^{k+1} - X'^k \quad (11)$$
$$= \eta' \nabla_{X'} L(X'^k; Y, w) + \sqrt{2\eta'} \varepsilon \mathcal{N}(0, \mathbb{I}),$$
$$X'^K - X'^{K-1}$$
$$= P\gamma \left( \eta' \nabla_{X'} L(X'^{K-1}; Y, w) + \sqrt{2\eta'} \varepsilon \mathcal{N}(0, \mathbb{I}) \right)$$

where element-wise projection $P_\gamma(z) = z$ if $|z| < 1/\gamma$ and $P_\gamma(z) = 1/\gamma$ if $|z| > 1/\gamma$. Empirically, we also explored an alternate implementation where the projection takes place in each inner iteration $k$, however, we find the version as described in Algorithm 2 (Appendix C) to give better results.

In both Algorithms 1 and 2, we initialize the Langevin update step with a random normal perturbation $\delta_i$ radius of benign samples, which is inversely proportional to $\gamma$.

## 3. Experiments

In this section we perform experiments on a five-layer convolutional model with 3 CNN and 2 fully connected layers, used in (Zhang et al., 2019b; Carlini & Wagner, 2017a),

*Table 1.* Robust percentage accuracies of 5-layer convolutional net for MNIST against $\ell_2, \epsilon = 2$ attack.

| Attack→ ↓ Defense | Benign Acc | $\ell_2$ PGD-40 | $\ell_2$ CW |
|---|---|---|---|
| SGD | **99.38** | 19.40 | 13.20 |
| Entropy SGD | 99.24 | 19.12 | 14.52 |
| $\ell_2$ PGD-AT | 98.76 | 72.94 | - |
| TRADES | 97.54 | **76.08** | - |
| MMA | **99.27** | 73.02 | 72.72 |
| $\ell_2$ ATENT | 98.66 | **77.21** | **76.72** |

*Table 2.* Robust accuracies (in percentages) of 5-layer convolutional net for MNIST against $\ell_\infty, \epsilon = 0.3$ attack.

| Attack→ ↓ Defense | Benign Acc | $\ell_\infty$ PGD-20 $\epsilon_\infty = 0.3$ | $\ell_\infty$ CW $\epsilon_\infty = 0.3$ |
|---|---|---|---|
| SGD | 99.39 | 0.97 | 32.37 |
| Entropy SGD | 99.24 | 1.17 | 34.34 |
| $\ell_\infty$ PGD-AT | 99.36 | 96.01 | 94.25 |
| TRADES | **99.48** | 96.07 | 94.03 |
| MMA | 98.92 | 95.25 | 94.77 |
| MART | 98.74 | **96.48** | **96.10** |
| $\ell_\infty$ ATENT | **99.45** | 96.44 | 97.40 |

trained on MNIST. We also train a WideResNet-34-10 on CIFAR10 (as used in (Zhang et al., 2019b)) as well as ResNet20. Due to space constraints, we present supplemental results in Appendix B. We conduct our experiments separately on networks specifically trained for $\ell_2$ attacks and those trained for $\ell_\infty$ attacks. We also test randomized smoothing for our $\ell_2$-ATENT model.

*Attacks:* For $\ell_2$ attacks, we test PGD-40 with 10 random restarts, and CW2 attacks at radius $\epsilon_2 = 2$ for MNIST and PGD-40 and CW2 attacks at $\epsilon_2 = 0.43 (\approx \epsilon_\infty = 2/255)$ and $\epsilon_2 = 0.5 = 128/255$ for CIFAR10. For $\ell_\infty$ attacks, we test PGD-20, $\ell_\infty$CW, DeepFool attacks at radiii $\epsilon_\infty = 0.3$ for MNIST and $\epsilon_\infty = 0.031 = 8/255$ for CIFAR10. We test ATENT at other attack radii in Appendix B. For implementing the attacks, we use the Foolbox library (Rauber et al., 2017) and the Adversarial Robustness Toolbox (Nicolae et al., 2018).

*Defenses:* We compare models trained using: SGD (vanilla), Entropy SGD (Chaudhari et al., 2019), PGD-AT (Madry et al., 2018) with random starts (or PGD-AT(E) with random start, early stopping (Rice et al., 2020)), TRADES (Zhang et al., 2019b), MMA (Ding et al., 2019) and MART (Wang et al., 2019). Wherever available, we use pretrained models to tabulate robust accuracy results for PGD-AT, TRADES, MMA and MART as presented in their published versions. Classifiers giving the best and second best accuracies are highlighted in each category.

*Smoothing:* We also test randomized smoothing (Cohen et al., 2019) in addition to our adversarial training to evaluate certified robust accuracies.

*MNIST:* In Tables 1 and 2, we tabulate the robust accuracy for 5-layer convolutional network trained using the various approaches discussed above for both $\ell_2$ and $\ell_\infty$ attacks respectively. Due to space constraints we push details of training setup to Appendix B.

*CIFAR10:* Next, we extend our experiments to CIFAR-10 using a WideResNet 34-10 as described in (Zhang et al., 2019b; Wang et al., 2019) as well as ResNet-20. Results for $\ell_\infty$ attacks are in Table 3 and $\ell_2$ attacks are in Table 3

*Table 3.* Robust accuracies of WRN34-10 net for CIFAR10 against $\ell_\infty$ attack of $\epsilon = 8/255$.

| Defense→ ↓ Attack | PGD AT | TRADES | MART | ATENT $\ell_\infty$ |
|---|---|---|---|---|
| Benign | **87.30** | 84.92 | 84.17 | **85.67** |
| $\ell_\infty$ PGD-20 (E) | 47.04 56.80 | 56.61 | **57.39** | 57.23 |
| $\ell_\infty$ CW | 49.27 | **62.67** | 54.53 | 62.34 |
| $\ell_\infty$ DeepFool | - | **58.15** | 55.89 | **57.21** |

in Appendix B. For PGD-AT (and PGD-AT (E)), TRADES, and MART, we use the default values stated in their corresponding papers.

***Importance of early stopping:*** Because WRN34-10 is highly overparameterized ($\approx 48$ million parameters), it tends to overfit adversarially-perturbed CIFAR10 examples. The success of TRADES (and also PGD) in (Rice et al., 2020) relies on an early stopping condition and corresponding learning rate scheduler. We strategically search different early stopping points and report the best possible robust accuracy from different stopping points.

We test efficiency of our $\ell_2$-based defense on both $\ell_2$ attacks, as well as compute $\ell_2$ certified robustness for the smoothed version of ATENT against smoothed TRADES (Blum et al., 2020) in Table 5 in Appendix B. We find that our formulation of $\ell_2$ ATENT is both robust against $\ell_2$ attacks, as well as gives a competitive certificate against adversarial perturbations for ResNet20 on CIFAR10.

In Appendix B we also consider a pre-trained WRN34-10 and fine tune it using ATENT, similar to the approach in (Jeddi et al., 2020). We find that ATENT can be used to fine tune a naturally pretrained model at lower computational complexity to give competitive robust accuracies while almost retaining the performance on benign data.

## Acknowledgments

This work was supported in part by NSF grants CCF-2005804 and CCF-1815101.

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

# A. Prior Work

Evidence for the existence of adversarial inputs for deep neural networks is by now well established (Carlini & Wagner, 2017b; Dathathri et al., 2017; Goodfellow et al., 2015; Goodfellow, 2018; Szegedy et al., 2013; Moosavi-Dezfooli et al., 2017). In image classification, the majority of attacks have focused on the setting where the adversary confounds the classifier by adding an imperceptible perturbation to a given input image. The range of the perturbation is pre-specified in terms of bounded pixel-space $\ell_p$-norm balls. Specifically, an $\ell_p$- attack model allows the adversary to search over the set of input perturbations $\Delta_{p,\epsilon} = \{\delta : \|\delta\|_p \leq \epsilon\}$ for $p = \{0, 1, 2, \infty\}$.

Initial attack methods, including the Fast Gradient Sign Method (FGSM) and its variants (Goodfellow et al., 2014a; Kurakin et al., 2016), proposed techniques for generating adversarial examples by ascending along the sign of the loss gradient:

$$x_{adv} = x + \epsilon \operatorname{sgn}(\nabla_x L(f(\hat{w}; x), y)),$$

where $(x_{adv} - x) \in \Delta_{\infty,\epsilon}$. Madry *et. al.* (Madry et al., 2018) proposed a stronger adversarial attack via projected gradient descent (PGD) by iterating FGSM several times, such that

$$x^{t+1} = \Pi_{x+\Delta_{p,\epsilon}}(x^t + \alpha \operatorname{sgn}(\nabla_x L(f(\hat{w}; x), y)),$$

where $p = \{2, \infty\}$. These attacks are (arguably) the most successful available attack techniques reported to date, and serve as the starting point for our comparisons. Both Deep Fool (Moosavi-Dezfooli et al., 2016) and Carlini-Wagner (Carlini & Wagner, 2017a) construct an attack by finding smallest possible perturbation that can flip the label of the network output.

Several strategies for defending against attacks have been developed. In (Madry et al., 2018), adversarial training is performed via the min-max formulation Eq. 1. The inner maximization is solved using PGD, while the outer objective is minimized using stochastic gradient descent (SGD) with respect to $w$. This can be slow to implement, and speed-ups have been proposed in (Shafahi et al., 2019) and (Wong et al., 2020). In (Li et al., 2018a; Cohen et al., 2019; Lecuyer et al., 2019; Salman et al., 2019b;a), the authors developed certified defense strategies via randomized smoothing. This approach consists of two stages: the first stage consists of training with noisy samples, and the second stage produces an ensemble-based inference. See (Ren et al., 2020) for a more thorough review of the literature on various attack and defense models.

Apart from minimizing the worst case loss, approaches which minimize the upper bound on worst case loss include (Wong & Kolter, 2018; Tjeng et al., 2018; Gowal et al., 2019). Another breed of approaches use a modified loss function which considers surrogate adversarial loss as an added regularization, where the surrogate is cross entropy (Zhang et al., 2019b) (TRADES), maximum margin cross entropy (Ding et al., 2019) (MMA) and KL divergence (Wang et al., 2019) (MART) between adversarial sample predictions and natural sample predictions.

In a different line of work, there have been efforts towards building neural network networks with improved generalization properties. In particular, heuristic experiments by (Hochreiter & Schmidhuber, 1997; Keskar et al., 2016; Li et al., 2018b) suggest that the loss surface at the final learned weights for well-generalizing models is relatively "flat" [1]. Building on this intuition, Chaudhari *et. al.* (Chaudhari et al., 2019) showed that by explicitly introducing a smoothing term (via entropic regularization) to the training objective, the learning procedure weights towards regions with flatter minima by design. Their approach, Entropy-SGD (or ESGD), is shown to induce better *generalization* properties in deep networks. We leverage this intuition, but develop a new algorithm for training deep networks with better *adversarial robustness* properties.

# B. Additional experiments and details

In this section, we provide additional details as well as experiments to supplement those in Section 3. All results were generated using an Intel(R) Xeon(R) W-2195 CPU 2.30GHz Lambda cluster with 18 cores and a NVIDIA TITAN GPU running PyTorch version 1.4.0.

## B.1. Detailed training setup

*Architectures:* For MNIST- $\ell_\infty$ experiments, we consider a CNN architecture with the following configuration (same as (Zhang et al., 2019b)). Feature extraction consists of the following sequence of operations: two layers of 2-D convolutions with 32 channels, kernal size 3, RelU activation each, followed by maxpooling by factor 2, followed by two layers of 2-D

---

[1]This is not strictly necessary, as demonstrated by good generalization at certain sharp minima (Dinh et al., 2017).

*Table 4.* Percentage robust accuracies of ResNet-20 for CIFAR10 against $\ell_2$ attack

| ↓ Algorithm/Attack→ | Model | Training param | Benign | PGD-10 $\epsilon_2 = 0.5$ | PGD-10 $\epsilon_2 = 1$ |
|---|---|---|---|---|---|
| PGD-AT | WideResNet28-4 | $\epsilon_2 = 1$ | 83.25 | 66.69 | 46.11 |
| MMA | WideResNet28-4 | $d = 1$ | 88.92 | 66.81 | 37.22 |
| $\ell_2$ ATENT | ResNet20 | $\gamma = 0.05, \varepsilon = 0.001\mathcal{N}(0, \mathbf{1})$ | 85.44 | 65.12 | 47.38 |
| | | | | | $\epsilon_2=0.435$ |
| TRADES (smooth) | ResNet20 | $\epsilon_2 = 0.435, \sigma = 0.12$ | 75.13 | | 61.03 |
| $\ell_2$ ATENT | ResNet20 | $\gamma = 0.05, \sqrt{2\eta'}\varepsilon = 0.12\mathcal{N}(0, \mathbf{1})$ | 72.10 | | 64.53 |

convolutions with 64 channels, kernel size 3, ReLU activation, and finally another maxpool (by 2) operation. This is followed by the classification module, consisting of a fully connected layer of size $1024 \times 200$, ReLU activation, dropout, another fully connected layer of size $200 \times 200$, ReLU activation and a final fully connected layer of size $200 \times 10$. Effectively this network has 4 convolutional and 3 fully connected layers. We use batch size of 128 with this configuration.

***Training setup for MNIST:*** Complete details are provided in Appendix B. Our experiments for $\ell_2$ attack are presented in Table 1. We perform these experiments on a LeNet5 model imported from the Advertorch toolbox (architecture details are provided in the supplement). For $\ell_2$-ATENT we use a batch-size of 50 and SGD with learning rate of $\eta = 0.001$ for updating weights. We set $\gamma = 0.05$ and noise $\varepsilon \sim 0.001\mathcal{N}(0, \mathbb{I})$. We perform $K = 40$ Langevin epochs and set the Langevin parameter $\alpha = 0.9$, and step $\eta' = 0.25$. For attack, we do a 40-step PGD attack with $\ell_2$-ball radius of $\epsilon = 2$. The step size for the PGD attack is 0.25, consistent with the configuration in (Ding et al., 2019). We perform early stopping by tracking robust accuracies of validation set and report the best accuracy found.

In Table 2, we use a SmallCNN configuration as described in (Zhang et al., 2019b) (architecture in supplement). We use a batch-size of 128, SGD optimizer with learning rate of $\eta = 0.01$ for updating weights. We set $\gamma = 3.33$ and noise $\varepsilon \sim 0.001\mathcal{N}(0, \mathbb{I})$. We perform $L = 40$ Langevin epochs and we set the Langevin parameter $\alpha = 0.9$, and step $\eta' = 0.01$, consistent with the configuration in (Zhang et al., 2019b). For the PGD attack, we use a 20-step PGD attack with step-size 0.01, for $\ell_\infty$-ball radius of $\epsilon = 0.3$. We perform an early stopping by tracking robust accuracies on the validation set and report the best accuracy found. Other attack configurations can be found in the supplement.

Our experiments on the Entropy-SGD (row 2 in Tables 1 and 2) trained network suggests that networks trained to find flat minima (with respect to weights) are not more robust to adversarial samples as compared to vanilla SGD.

For MNIST-$\ell_2$ experiments, we consider the LeNet5 model from the Advertorch library (same as (Ding et al., 2019)). This consists of a feature extractor of the form - two layers of 2-D convolutions, first one with 32 and second one with 64 channels, ReLU activation and maxpool by factor 2. The classifier consists of one fully connected layer of dimension $3136 \times 1024$ followed by ReLU activation, and finally another fully connected layer of size $1024 \times 10$. We use batch size of 50 with this configuration.

***Training setup for CIFAR10:*** Complete details in Appendix B. Robust accuracies of WRN-34-10 classifer trained using state of art defense models are evaluated at the $\ell_\infty$ attack benchmark requirement of radius $\epsilon = 8/255$, on CIFAR10 dataset and tabulated in Table 3. For $\ell_\infty$-ATENT, we use a batch-size of 128, SGD optimizer for weights, with learning rate $\eta = 0.1$ (decayed to 0.01 at epoch 76), 76 total epochs, weight decay of $5 \times 10^{-4}$ and momentum 0.9. We set $\gamma = 1/(0.0031)$, $K = 10$ Langevin iterations, $\varepsilon = 0.001\mathcal{N}(0, \mathbb{I})$, at step size $\eta' = 0.007$. We test against 20-step PGD attack, with step size 0.003, as well as $\ell_\infty$-CW and Deep Fool attacks using FoolBox. $\ell_\infty$-ATENT is consistently among the top two performers at benchmark configurations.

For CIFAR-$\ell_\infty$ experiments we consider a WideResNet with 34 layers and widening factor 10 (same as (Zhang et al., 2019b) and (Madry et al., 2018)). It consists of a 2-D convolutional operation, followed by 3 building blocks of WideResNet, ReLU, 2D average pooling and fully connected layer. Each building block of the WideResNet consists of 5 successive operations of batch normalization, ReLU, 2D convolution, another batch normalization, ReLU, dropout, a 2-D convolution and shortcut connection. We use batch size of 128 with this configuration.

For CIFAR-$\ell_2$ experiments, we consider a ResNet with 20 layers. This ResNet consists of a 2-D convolution, followed by three blocks, each consisting of 3 basic blocks with 2 convolutional layers, batch normalization and ReLU. This is finally followed by average pooling and a fully connected layer. We use batch size of 256 with this configuration.

*Table 5.* Smoothed robust accuracies for CIFAR10 against $\ell_\infty$ attack of $\epsilon = 2/255$ ($\ell_2, \epsilon = 0.435$), smoothing factor $\sigma = 0.12$.

| Smoothing radius → 
 ↓ Defense | ResNet 
 Type | Standard 
 0 | $\epsilon_\infty$ 
 2/255 |
|---|---|---|---|
| Crown IBP (Zhang et al., 2019a) | 110 | 72.0 | 54.0 |
| Smoothing (Wong et al., 2018) | 110 | 68.3 | 53.9 |
| SmoothAdv (Salman et al., 2019b) | 110 | **82.1** | **60.8** |
| TRADES Smoothing (Blum et al., 2020) | 110 | **78.7** | **62.6** |
| TRADES Smoothing | 20 | **78.2** | **58.1** |
| ATENT (ours) | 20 | **72.2** | **55.41** |

***Training SGD and Entropy SGD models for MNIST experiments:*** For SGD, we trained the 7-layer convolutional network setup in (Zhang et al., 2019b; Carlini & Wagner, 2017a) with the MNIST dataset, setting batch size of 128, for $\ell_\infty$ SGD optimizer using a learning rate of 0.1, for 50 epochs. For Entropy SGD, with 5 langevin steps, and $\gamma = 10^{-3}$, batch size of 128 and learning rate of 0.1 and 50 total epochs.

## B.2. $\ell_2$ ATENT

**$\ell_2$-*PGD attacks on CIFAR10:*** We explore the effectiveness of $\ell_2$-ATENT as a defense against $\ell_2$ perturbations. These results are tabulated in Table 4. We test 10-step PGD adversarial attacks at $\epsilon_2 = 0.5$ and $\epsilon_2 = 1$. For the purpose of this comparison, we compare pretrained models of MMA and PGD-AT at $\epsilon_2 = 1$. To train ATENT, we use $\gamma = 0.08$ for $\epsilon_2 = 1$ 10 step attack (with $2.5\epsilon_2/10$ step size), $K = 10$ langevin iterations, langevin step $\eta' = 2\epsilon_2/K$, learning rate for weights $\eta = 0.1$. We also compare models primarily trained to boost the certificate of randomized smoothing. For this we train a ResNet20 model for both TRADES (at default parameter setting) and $\ell_2$ ATENT, at $\eta' =$ and $\gamma =$, such that the effective noise standard deviation is 0.12. These models are tested against PGD-10 attacks at radius $\epsilon_2 = 0.435$. In all $\ell_2$ ATENT experiments, we choose the value of $\gamma$ such that the perturbation $\|X'^K - X\|_F \approx \epsilon_2$ of corresponding models of TRADES and PGD-AT. For all ATENT experiments, we set $\alpha = 0.9$.

***Experiments on randomized smoothing:*** Since the formulation of ATENT is similar to a noisy PGD adversarial training algorithm, we test its efficiency towards randomized smoothing and producing a higher robustness certificate (Table 5). For this we train a ResNet-20 on CIFAR10, at $\gamma = 0.05, \eta' = 0.02, \eta = 0.1, K = 10$, and tune the noise $\varepsilon$, such that effective noise $\sqrt{2\eta'}\varepsilon$ has standard deviation $\sigma = 0.12$. We compare the results of randomized smoothing to established benchmarks on ResNet-110 (results have been borrowed from Table 1 of (Blum et al., 2020)). as well as a smaller ResNet-20 model trained using TRADES at its default settings. We observe that without any modification to the current form of ATENT, our method is capable of producing a competitive certificate to state of art methods. In future work we aim to design modifications to ATENT which can serve the objective of certification.

## B.3. $\ell_\infty$ ATENT

***Training characteristics of $\ell_\infty$ ATENT:*** In Figure 2 we display the training curves of ATENT. As shown, the robust accuracies spike sharply after the first learning rate decay, followed by an immediate decrease in robust accuracies. This behavior is similar to that observed in (Rice et al., 2020). This is also the key intuition used in the design of the learning rate scheduler for TRADES.

***ATENT as Attack:*** For our $\ell_\infty$-ATENT WideResNet-34-10, we also test $\ell_\infty$-ATENT as an attack. We keep the same configuration as that of PGD-20, for ATENT. We compare the performance of our $\ell_\infty$-ATENT trained model (specifically designed to work against $\epsilon_\infty$=8/255 attacks). The values (Table 6) suggest that the adversarial perturbations generated by

*Table 6.* Percentage robust accuracies for CIFAR10 against $\ell_\infty$ PGD and ATENT attacks of different radii.

| Attack radius → 
 ↓ Attack | 2/255 | 4/255 | 8/255 | 12/255 |
|---|---|---|---|---|
| PGD-20 | 79.83 | 73.35 | 57.23 | 39.37 |
| ATENT-20 | 79.95 | 73.76 | 59.69 | 47.53 |

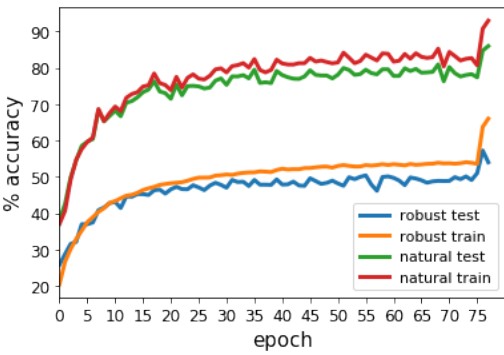

*Figure 2.* Benign training, test and robust training. test accuracies of ATENT. The learning rate is decayed at epoch 76, where the robust test accuracy peaks. This is the accuracy reported.

---

**Algorithm 1** $\ell_2$-ATENT

1: **Input:** $X = [X_{B_1}, X_{B_2} \ldots X_{B_J}], f, \eta, \eta', w = w^0, \gamma, \varepsilon, \alpha$
2: **for** $t = 1, \cdots T$ **do**
    (outer loop of SGD)
3:    **for** $j = 1, \cdots J$ **do**
    (scan through all batches of data)
4:     $x_i^0 \leftarrow x_i + \delta_i$ $\{\forall x_i \in X_{B_j}, K$ is number of samples generated using Langevin dynamics$\}$
5:     $\mu^j \leftarrow 0$
6:     **for** $k = 1, \cdots, K$ **do**
7:        $dx'^k \leftarrow \frac{1}{n_j} \sum_{i=1}^{n_j} \nabla_{x=x'^k} L(f(w^t; x)) + \gamma(x^k - x'^k)$
8:        $x'^{k+1} \leftarrow x'^k + \eta' dx'^k + \sqrt{2\eta'} \varepsilon \mathcal{N}(0,1)$ {Langevin update}
9:        $\mu^k \leftarrow \frac{1}{B} \sum_{x_i \in X_{B_j}} L(w^t; x'^{k+1})$ {augmented batch loss for $X_{B_j}$}
10:      $\mu^j \leftarrow (1-\alpha)\mu^j + \alpha\mu^k$
11:     **end for**
12:     $dL^t \leftarrow \nabla_w \mu^j$
13:     $w^{t+1} \leftarrow w^t - \eta dL^t$
14:    **end for**
15: **end for**
16: **Output** $\hat{w} \leftarrow w^T$

---

ATENT are similar in strength to those produced by PGD (worst possible attack).

***Computational complexity:*** In terms of computational complexity, ATENT matches that of PGD and TRADES, as can be observed from the fact that all three approaches are nested iterative optimizations. Due to the high computational complexity of all adversarial algorithms, we test a fine-tuning approach, to trade computational complexity for accuracy. This method is suggested in (Jeddi et al., 2020). In this context, we take a pre-trained WideResNet-34-10 which has been trained on benign CIFAR10 samples only. This model is then fine tuned on adversarial training data, via $\ell_\infty$ ATENT using a low learning rate $\eta = 0.0001$ and trained for only 20 epochs. The final robust accuracy at $\epsilon_\infty = 8/255$ is 52.1%. This is accuracy marginally improves upon the robust accuracy observed (51.7%) for fine-tuned WideResNet-28-10 PGD-AT trained model in (Jeddi et al., 2020). This experiments suggests that ATENT is amenable for fine tuning pretrained benign models using lesser computation, but at the cost of slightly reduced robust accuracy (roughly 5% drop at benchmark of $\epsilon_\infty = 8/255$).

## C. Algorithm, Proofs and Derivations

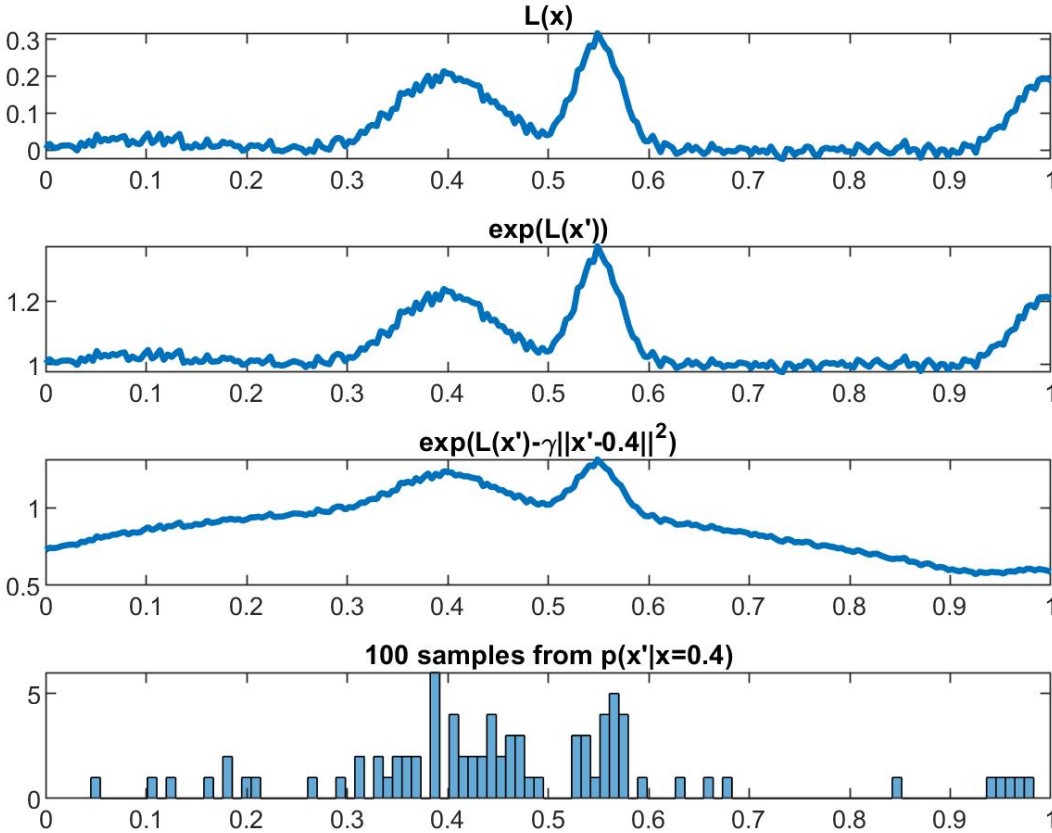

*Figure 3.* Illustration of the sampling procedure in Assumption 1 at fixed weights $w$. The distribution produces samples $x'$ from distribution $p(x'|x = 0.4)$, and we compute the average loss over these samples. Effectively, this encourages ATENT to search for $w$ where $L(x; w)$ is relatively flat in the neighborhood of $x$.

---

**Algorithm 2** $\ell_\infty$-ATENT

---

1: **Input:** $X = [X_{B_1}, X_{B_2} \ldots X_{B_J}], f, \eta, \eta', w = w^0, \gamma, \varepsilon, \alpha$
2: **for** $t = 1, \cdots T$ **do**
     (outer loop of SGD)
3:    **for** $j = 1, \cdots J$ **do**
       (scan through all batches of data)
4:       $x_i^0 \leftarrow x_i + \delta_i \; \{\forall x_i \in X_{B_j}, K$ is number of samples generated using Langevin dynamics$\}$
5:       $\mu^j \leftarrow 0$
6:       **for** $k = 1, \cdots, K$ **do**
7:          $dx'^k \leftarrow \frac{1}{n_j} \sum_{i=1}^{n_j} \nabla_{x=x'^k} L(f(w^t; x))$
8:          $x'^{k+1} \leftarrow x'^k + P_\gamma^K(\eta' dx'^k + \sqrt{2\eta'}\varepsilon \mathcal{N}(0,1))$ {update follows Eq.11, projection active in $K^{th}$ iteration only.}
9:          $\mu^k \leftarrow \frac{1}{B} \sum_{x_i \in X_{B_j}} L(w^t; x'^{k+1})$ {augmented batch loss for $X_{B_j}$}
10:        $\mu^j \leftarrow (1 - \alpha)\mu^j + \alpha\mu^k$
11:       **end for**
12:       $dL^t \leftarrow \nabla_w \mu^j$
13:       $w^{t+1} \leftarrow w^t - \eta dL^t$
14:    **end for**
15: **end for**
16: **Output** $\hat{w} \leftarrow w^T$

---

## C.1. Theoretical properties of the augmented loss

We now state an informal theorem on the conditions required for convergence of SGLD in Eq. 7 for estimating adversarial samples $X'$.

**Lemma C.1.** *The effective loss $F(X'; X, Y, w) := \frac{\gamma}{2}\|X - X'\|_F^2 - \mathcal{L}(X'; Y, w)$ which guides the Langevin sampling process in Eq. 8 is*

1. *$\beta + \gamma$ smooth if $\mathcal{L}(X; Y, w)$ is $\beta$-smooth in $X$.*

2. *$\left(\frac{\gamma}{4}, \frac{L^2}{\gamma} + \frac{\gamma}{2}\|X\|_F^2\right)$ dissipative if $\mathcal{L}(X; Y, w)$ is $L$-Lipschitz in $X$.*

One can then use smoothness and dissipativity of $F(X'; Y, w)$ to show convergence of SGLD for the optimization over $X'$ (Eq. 8) via Theorem 3.3 of (Xu et al., 2017).

We first derive smoothness conditions for the effective loss

$$F(X'; X, Y, w) := \frac{\gamma}{2}\|X - X'\|_F^2 - \mathcal{L}(X'; Y, w), \quad \forall X_1', X_2'.$$

We use abbreviations $p(X') := p(X'; X, Y, w), F(X') := F(X'; X, Y, w), \mathcal{L}(X'; Y, z) := \mathcal{L}(X')$ and $\mathcal{L}(X; Y, z) := \mathcal{L}(X)$, and assume that $X$ and $X'$ are vectorized. Unless specified otherwise, $\|\cdot\|$ refers to the vector 2-norm.

*Proof.* Let us show that $\|\nabla_{X'} F(X_2') - \nabla_{X'} F(X_1')\| \leq \beta'\|X_2' - X_1'\|$. If the original loss function is $\beta$ smooth, i.e.,

$$\|\nabla_{X'}\mathcal{L}(X_2') - \nabla_{X'}\mathcal{L}(X_2')\| \leq \beta\|X_2' - X_1'\|,$$

then:

$$\begin{aligned}
\|\nabla_{X'} F(X_2') - \nabla_{X'} F(X_1')\| &\leq \| -\nabla_{X'}\mathcal{L}(X_2') + \nabla_{X'}\mathcal{L}(X_1') - \gamma(X - X_2') + \gamma(X - X_1')\| \\
&\leq \|\nabla_{X'}\mathcal{L}(X_2') - \nabla_{X'}\mathcal{L}(X_1')\| + \|\gamma(X_2' - X_1')\| \\
&\leq (\beta + \gamma)\|X_2' - X_1'\|
\end{aligned}$$

by application of the triangle inequality.

Next, we establish conditions required to show $(m, b)$-dissipativity for $F(X')$, i.e. $\langle \nabla_{X'} F(X'), X' \rangle \geq m\|X'\|_2^2 - b$ for positive constants $m, b > 0$, $\forall X'$. To show that:

$$\langle \nabla_{X'} F(X'), X' \rangle \geq m\|X'\|_2^2 - b$$

where the left side of inequality can be expanded as:

$$
\begin{aligned}
\langle \nabla_{X'} F(X'), X' \rangle &= \langle -\nabla_{X'} \mathcal{L}(X') + \gamma(X' - X), X' \rangle \\
&= \langle -\nabla_{X'} \mathcal{L}(X'), X' \rangle + \gamma\|X'\|_2^2 - \gamma\langle X, X' \rangle \\
&= \langle -\nabla_{X'} \mathcal{L}(X'), X' \rangle + \gamma\|X'\|_2^2 - \frac{\gamma}{2}\left(\|X'\|_2^2 + \|X\|_2^2 - \|X - X'\|_2^2\right) \\
&\geq \langle -\nabla_{X'} \mathcal{L}(X'), X' \rangle + \gamma\|X'\|_2^2 - \frac{\gamma}{2}\left(\|X'\|_2^2 + \|X\|_2^2\right) \\
&= \langle -\nabla_{X'} \mathcal{L}(X'), X' \rangle + \frac{\gamma}{2}\|X'\|_2^2 - \frac{\gamma}{2}\|X\|_2^2
\end{aligned}
\tag{12}
$$

To find the inner product $\langle -\nabla_{X'} \mathcal{L}(X'), X' \rangle$, we expand squares:

$$\|\nabla_{X'} \mathcal{L}(X') - \frac{\gamma}{2} X'\|^2 = \|\nabla_{X'} \mathcal{L}(X')\|_2^2 + \frac{\gamma^2}{4}\|X'\|_2^2 - \gamma\langle \nabla_{X'} \mathcal{L}(X'), X' \rangle \geq 0$$

$$-\langle \nabla_{X'} \mathcal{L}(X'), X' \rangle \geq -\frac{\|\nabla_{X'} \mathcal{L}(X')\|_2^2}{\gamma} - \frac{\gamma}{4}\|X'\|_2^2$$

Plugging this into (12), and assuming Lipschitz continuity of original loss $\mathcal{L}(X')$, i.e., $\|\nabla_{X'} \mathcal{L}(X')\|_2 \leq L$:

$$
\begin{aligned}
\langle \nabla_{X'} F(X'), X' \rangle &\geq \langle -\nabla_{X'} \mathcal{L}(X'), X' \rangle + \frac{\gamma}{2}\|X'\|_2^2 - \frac{\gamma}{2}\|X\|_2^2 \\
&\geq -\frac{\|\nabla_{X'} \mathcal{L}(X')\|_2^2}{\gamma} - \frac{\gamma}{4}\|X'\|_2^2 + \frac{\gamma}{2}\|X'\|_2^2 - \frac{\gamma}{2}\|X\|_2^2 \\
&= \frac{\gamma}{4}\|X'\|_2^2 - \left(\frac{L^2}{\gamma} + \frac{\gamma}{2}\|X\|_2^2\right) \\
&= m\|X'\|_2^2 - b
\end{aligned}
$$

where $m = \frac{\gamma}{4}$ and $b = \frac{L^2}{\gamma} + \frac{\gamma}{2}\|X\|_2^2$. Thus, $F(X')$ is $(\frac{\gamma}{4}, \frac{L^2}{\gamma} + \frac{\gamma}{2}\|X\|_2^2)$ dissipative, if $\mathcal{L}(X')$ is $L$-Lipschitz.

$\square$

With Lemma C.1 we can show convergence of the SGLD inner optimization loop. To minimize overall loss function, the data entropy loss $\mathcal{L}_{DE}$ is minimized w.r.t. $w$, via Stochastic Gradient Descent (SGD). The gradient update for weights $w$ are designed via (5) as follows:

$$
\begin{aligned}
\nabla_w \mathcal{L}_{DE}(w; X, Y, \gamma) &= \nabla_w \int_{X'} \mathcal{L}(X'; Y, w) p(X'; X, Y, w, \gamma) dX' = \nabla_w \mathbb{E}_{X' \sim p(X'; X, Y, w, \gamma)}[\mathcal{L}(X'; Y, w)] \\
&= \int_{X'} \nabla_w \left(\mathcal{L}(X'; Y, w) p(X'; X, Y, w, \gamma)\right) dX' \\
&= \int_{X'} \nabla_w \mathcal{L}(X'; Y, w) \cdot p(X'; X, Y, w, \gamma) + \nabla_w \mathcal{L}(X'; Y, w) \cdot \mathcal{L}(X'; Y, w) \cdot p(X'; X, Y, w, \gamma) dX' \\
&= \int_{X'} \nabla_w \mathcal{L}(X'; Y, w) \cdot \left(\mathcal{L}(X'; Y, w) + 1\right) \cdot p(X'; X, Y, w, \gamma) dX' \\
&= \mathbb{E}_{X' \sim p(X'; X, Y, w, \gamma)} \left(\nabla_w \mathcal{L}(X'; Y, w) \cdot \left(\mathcal{L}(X'; Y, w) + 1\right)\right)
\end{aligned}
$$

Then a loose upper bound on Lipschitz continuity of $\mathcal{L}_{DE}$ is $\|\nabla_w \mathcal{L}_{DE}(w; X, Y, \gamma)\|_2 \leq \bar{L}(R + 1)$, if original loss is $\bar{L}$-Lipschitz in $w$ and $\mathcal{L}(X) \leq R$. Due to the complicated form of this expression, establishing $\beta$-smoothness will require extra rigor. We push a more thorough evaluation of the convergence of the outer SGD loop to future work.

---

**Algorithm 3** Entropy SGD

---

1: **Input:** $X = [X_{B_1}, X_{B_2} \dots X_{B_J}], f, \eta, \eta', w = w^0, \gamma, \alpha, \varepsilon$
2: **for** $t = 0, \cdots T - 1$ **do**
3:     **for** $j = 1, \cdots J$ **do**
4:       $w'^0 \leftarrow w^t, \mu^0 \leftarrow w^t$        {Repeat inner loop for all training batches $j$}
5:       **for** $k = 0, \cdots, K - 1$ **do**
6:         $dw'^k \leftarrow \frac{1}{n_j} \sum_{i=1}^{n_j} -\nabla_{w=w^k} L(f(w; x_i)) + \gamma(w^k - w'^k)$ {$\forall x_i \in X_{B_j}$}
7:         $w'^{k+1} \leftarrow w'^k + \eta' dw'^k + \sqrt{2\eta'} \varepsilon \mathcal{N}(0, 1)$ {Langevin update}
8:         $\mu^k \leftarrow (1 - \alpha)\mu^k + \alpha w'^{k+1}$
9:       **end for**
10:      $\mu^t \leftarrow \mu^K$
11:      $w^{t+1} \leftarrow w^t - \eta\gamma(w^t - \mu^t)$ {Repeat outer loop step for all training batches $j$}
12:     **end for**
13: **end for**
14: **Output** $\hat{w} \leftarrow w^T$

---

## C.2. Entropy SGD

In (Chaudhari et al., 2019) authors claim that neural networks that favor wide local minima have better generalization properties, in terms of perturbations to data, weights as well as activations. Mathematically, the formulation in Entropy SGD can be summarized as follows. A basic way to model the distribution of the weights of the neural network is using a Gibbs distribution of the form:

$$p(w; X, Y, \beta) = Z_{X,\beta}^{-1} \exp\left(-\beta \mathcal{L}(w; X, Y)\right)$$

When $\beta \to \infty$, this distribution concentrates at the global (if unique) minimizer of $\mathcal{L}(w^*; X, Y)$. A modified Gibbs distribution, with an additional smoothing parameter is introduced, which assumes the form:

$$p(w'; w, X, Y, \beta = 1, \gamma) = Z_{w,X,\gamma}^{-1} \exp\left(-\mathcal{L}(w'; X, Y) - \frac{\gamma}{2}\|w' - w\|_2^2\right) \tag{13}$$

where $Z_{w,X,\gamma}$ normalizes the probability.

Here $\gamma$ controls the width of the valley; if $\gamma \to \infty$, the sampling is sharp, and this corresponds to no smoothing effect, meanwhile $\gamma \to 0$ corresponds to a uniform contribution from all points in the loss manifold. The standard objective is:

$$\min_w \mathcal{L}(w; X, Y) := \min_w -\log\left(\exp\left(-\mathcal{L}(w; X, Y)\right)\right)$$
$$= \min_w -\log\left(\int_{w'} \exp\left(-\mathcal{L}(w'; X, Y)\right)\delta(w - w')dw'\right)$$

which can be seen as a sharp sampling of the loss function. Now, if one defined the Local Entropy as:

$$\mathcal{L}_{ent}(w; X, Y) = -\log(Z_{w,X,Y,\gamma})$$
$$= -\log\left(\int_{w'} \exp\left(-\mathcal{L}(w'; X, Y) - \frac{\gamma}{2}\|w - w'\|_2^2\right)dw'\right)$$

our new objective is to minimize this augmented objective function $\mathcal{L}_{ent}(w; X, Y)$, which resembles a smoothed version of

the loss function with a Gaussian kernel. The SGD update can be designed as follows:

$$
\begin{aligned}
\nabla_w \mathcal{L}_{ent}(w; X, Y) &= -\nabla_w(\log(Z_{w,X,Y,\gamma})) \\
&= Z_{w,X,\gamma}^{-1} \nabla_w(Z_{w,X,\gamma}) \\
&= Z_{w,X,\gamma}^{-1} \left( \int_{w'} \exp\left( -\mathcal{L}(w'; X, Y) - \frac{\gamma}{2} \|w - w'\|_2^2 \right) \cdot \gamma(w - w')dw' \right) \\
&= \int_{w'} p(w'; w, X, Y, \gamma) \cdot \gamma(w - w')dw' \\
&= \mathbb{E}_{w' \sim p(w')} \left[ \gamma(w - w') \right]
\end{aligned}
$$

Then, using this gradient, the SGD update for a given batch is designed as:

$$
w^+ = w - \eta \nabla_w \mathcal{L}_{ent}(w; X, Y)
$$

This gradient ideally requires computation over the entire training set at once; however can be extended to a batch-wise update rule by borrowing key findings from (Welling & Teh, 2011). This expectation for the full gradient is computationally intractable, however, Euler discretization of Langevin Stochastic Differential Equation, it can be approximated fairly well as

$$
w'^{t+1} = w'^t + \eta^t \nabla_{w'} \log p(w'^t) + \sqrt{2\eta} \mathcal{N}(0, \mathbb{I})
$$

such that after large enough amount of iterations $w^+ \to w^\infty$ then $w^\infty \sim p(w')$. One can estimate $\mathbb{E}_{w' \sim p(w')} \left[ \gamma(w - w') \right]$ by averaging over many such iterates from this process. This result is stated as it is from (Chaudhari et al., 2019): $\mathbb{E}_{w' \sim p(w')}[g(w')] = \frac{\sum_t \eta_t g(w'_t)}{\sum_t \eta_t}$. This leads to the algorithm shown in Algorithm 3. One can further accrue exponentially decaying weighted averaging of $g(w'_t)$ to estimate $\mathbb{E}_{w' \sim p(w')}[g(w')]$. This entire procedure is described in Algorithm 3.

This algorithm is then further guaranteed to find wide minima neighborhoods of $w$ by design, as sketched out by the proofs in (Chaudhari et al., 2019).

## C.3. Stochastic Gradient Langevin Dynamics

Stochastic Gradient Langevin Dynamics combines techniques of Stochastic Gradient Descent and Langevin Dynamics (Welling & Teh, 2011). Given a probability distribution $\pi = p(\theta; X)$, the following update rule allows us to sample from the distribution $\pi$:

$$
\theta^{t+1} = \theta^t + \eta \nabla_\theta p(\theta^t; X) + \sqrt{2\eta'} \varepsilon \tag{14}
$$

where $\eta'$ is step size and $\varepsilon$ is normally distributed. Then, as $t \to \infty$, $\theta \sim \pi$.

While this update rule in itself suffices, if the parameters are conditioned on a a training sample set $X$, which is typically large, the gradient term in Eq. 14 is expensive to compute. (Welling & Teh, 2011) shows that the following batch-wise update rule:

$$
\theta^{t+1} = \theta^t + \eta \nabla_\theta p(\theta^t; X_{B_j}) + \sqrt{2\eta'} \varepsilon
$$

suffices to produce a good approximation to the samples $\theta^{t \to \infty} \sim \pi$. In Algorithm 1, $\theta$ is the set of perturbed points $X'$. In each internal iteration, we look at subset of trainable parameters $X'_{B_j}$. We update the estimate for $X'_{B_j}$ by only considering data-points $X_{B_j}$ at a time. In the current formulation the set of iterable parameters $X'_{B_j}$ only 'see' a single batch of data $X_{B_j}$; a better estimate would require $X'_{B_j}$ to be updated by iteratively over all possible batches $X_{B_k}$, $k = 1, 2....J$. However in practice we observe that just using the corresponding batch $X_{B_{k=j}}$ suffices. In future work, we will explore the theoretical implications of this algorithmic design.

## C.4. PGD-Adversarial Training

In Algorithm 4 we describe the PGD-AT algorithm. In (Madry et al., 2018) authors demonstrate that PGD based-attack is the best possible attack that can be given for a given network and dataset combination. Theoretically,

$$
\bar{x}_{\text{worst}} = \arg \max_{\delta \in \Delta_p} L(f(\hat{w}; x + \delta), y) \tag{15}
$$

---

**Algorithm 4** PGD AT

---

1: **Input:** $[X_{B_1}, X_{B_2} \ldots X_{B_J}], f, \eta, \eta', w = w^0, \epsilon$
2: **for** $t = 0, \cdots T - 1$ **do**
3:   **for** $j = 1, \cdots J$ **do**
4:     $x'^0 \leftarrow x \; \{\forall x \in X_{B_j}\}$
5:     **for** $k = 0, \cdots, K - 1$ **do**
6:       $dx'^k \leftarrow \frac{1}{n} \sum_{i=1}^{n} \nabla_{x=x^k} L(f(w^t; x))$
7:       $x'^{k+1} \leftarrow x'^k + \eta' dx'^k$ {Gradient ascent}
8:       $\text{project}_{x+\Delta}(x', \epsilon)$
9:     **end for**
10:    $\mu^t \leftarrow L(w^t, x'^K)$ {batch loss for $X_{B_j}$}
11:    $dL^t \leftarrow \nabla_w \mu^t$ {gradient of batch loss}
12:    $w^{t+1} \leftarrow w^t - \eta dL^t$
13:   **end for**
14: **end for**
15: **Output** $\hat{w} \leftarrow w^T$

---

and if this maximization can be solved tractably, then a network trained with the following min-max formulation is said to be robust:

$$\min_w \max_{\delta \in \Delta_p} \mathcal{L}(f(\hat{w}; X + \delta, Y), Y) \tag{16}$$

Furthermore they show that first order based gradient approaches, such as SGD, are sufficient to suitably optimize the inner maximization over the perturbed dataset. This can be obtained using the following gradient *ascent* update rule:

$$X' = X' + \eta' \nabla_{X' \in X+\delta} \mathcal{L}(f(w, X' + \delta; Y), Y)$$

(see also Step 7 of Algorithm 4). Note that when $\Delta_p = \Delta_2$, this projection rule represents a noise-less version of the update rule in ATENT (see Algorithm 1, line 8).

Iterative Fast Gradient Sign (IFGS) method effectively captures a similar projection based approach which performs an update within an $\ell_\infty$ ball. This update is given by:

$$X' = X' + \eta' \text{sign}(\nabla_{X'} \mathcal{L}(f(w, X' + \delta; Y), Y))$$

Note that this update rule constructs an adversarial example within $\ell_\infty$-ball, during the training procedure. Meanwhile, given our proposed adversarial example sampling criterion in Assumption 2, our update rule is slightly different (see also Eq. 10).

### C.5. Comparison to PGD Adversarial Training

The updates of PGD-AT are similar to that of Algorithm 1, consisting broadly of two types of gradient operations in an alternating fashion - (i) an (inner) gradient with respect to samples $X$ (or batch-wise samples $X_{B_j}$) and (ii) an (outer) gradient with respect to weights $w$. While PGD-AT minimizes the *worst-case* loss in an $\epsilon$-neighborhood (specifically $\ell_2$ or $\ell_\infty$ ball) of $X$, ATENT minimizes an *average loss* over our specifically designed probability distribution (Assumption 2) in the neighborhood of $X$. Note that the gradient operation in Eq. 8 is also the gradient for the regularized version of inner maximation of the adversarial training problem (Madry et al., 2018), but with added noise term,

$$\max_{X'} \mathcal{L}(X'; X, Y, w) \quad s.t. \quad \|X' - X\|_F^2 \le \epsilon$$
$$\Leftrightarrow \max_{X'} \mathcal{L}(X'; X, Y, w) - \frac{\gamma}{2} \|X' - X\|_F^2 \tag{17}$$

constraint being satisfied if $\|X' - X\|_F$ is minimized, or $-\|X' - X\|_F$ is maximized).

The width of the Gaussian smoothing is adjusted with $\gamma$, which is analogous to controlling the projection radius $\epsilon$ in the inner-maximization of PGD-AT. Then the second and third terms in Eq. 8 are simply gradient of an $\ell_2$-regularization term over data space $X'$ and noise. In this way, ATENT can be re-interpreted as a stochastic formalization of $\ell_2$-PGD-AT, with noisy controlled updates.

## C.6. Comparison to randomized smoothing

In (Cohen et al., 2019), authors describe a defense to adversarial perturbations, in the form of smoothing. A smoothed classifier $g$, under isotropic Gaussian noise $\varepsilon = \mathcal{N}(0, \sigma^2 \mathbb{I})$, produces an output:

$$g(x) = \arg\max_{j} \mathbb{P}(f(x) + \varepsilon = j). \tag{18}$$

where $\mathbb{P}$ denotes probability distribution (see Appendix B for detailed discussion). SmoothAdv (Salman et al., 2019a) is an adversarial attack as well as defense for smoothed classifiers, which replaces standard loss with cross entropy loss of a smoothed classier. In comparison, we compute a smoothed version of the cross entropy loss of a standard classifier. This is similar to the setup of (Blum et al., 2020) (TRADES with smoothing). The procedure in Algorithm1 is therefore also amenable to randomized smoothing in its evaluation.