# OpenReview forum: "Adversarially Robust Learning via Entropic Regularization"
_ICML.cc/2021/Workshop/AML — ICML 2021 Workshop AML Poster_

### Official Review · Reviewer_df6v · 2021-06-20
**The authors provide a generally good augmented loss function for NN training leveraging adjacent adversarial examples.**

**Rating:** Accept
**Confidence:** 4

**Review:**

- The authors provide an augmented loss function for robust training of neural network, which considering adversarial samples with high loss in the original data point x's neighborhood. The authors also propose algorithms (ATENT) for optimizing the new augmented loss function with practical sampling methods and report robustness performance improvement over MINIST and CIFAR10 than baselines.
- However, ATENT doesn't always overperform all the baselines on robust accuracies (e.g. Table 2.), which leads to a question that whether all the approximation used for practical sampling might compromise the effectiveness of the augmented loss function. Besides, although the authors provide some theoretical analysis of proposed loss function, assuming the distribution of possible perturbations still seems to be a little arbitrary. But generally speaking, this is still a quite complete work with clear structure.

---

### Decision · Program_Chairs · 2021-06-21

**Decision:**

Accept (Poster)

**Comment:**

This paper proposed to learn adversarially robust models via entropic regularizations. The authors can further address the reviewer's comments.